# A Comparison of Slow Infusion Intermittent Feeding versus Gravity Feeding in Preterm Infants: A Randomized Controlled Trial

**DOI:** 10.3390/children10081389

**Published:** 2023-08-15

**Authors:** Funda Yavanoglu Atay, Ozlem Bozkurt, Suzan Sahin, Duygu Bidev, Fatma Nur Sari, Nurdan Uras

**Affiliations:** 1Department of Pediatrics, Division of Neonatology, Umraniye Training and Research Hospital, 34764 Istanbul, Türkiye; 2Department of Pediatrics, Division of Neonatology, Faculty of Medicine, Kocaeli University, 41380 Izmit, Türkiye; 3Department of Pediatrics, Division of Neonatology, Buca Seyfi Demirsoy Training and Research Hospital, Izmir Democracy University, 35140 Izmir, Türkiye; 4Neonatal Care Intensive Unit, Koru Sincan Hospital, 06934 Ankara, Türkiye; 5Department of Pediatrics, Division of Neonatology, Ankara City Hospital, 06800 Ankara, Türkiye; 6Department of Pediatrics, Division of Neonatology, Faculty of Medicine, Istinye University, 34517 Istanbul, Türkiye

**Keywords:** feeding intolerance, feeding methods, full enteral feeding, preterm

## Abstract

Background: The transition to full enteral feeding is important for ensuring adequate growth in preterm infants. Aims: The aim of this study was to investigate the effects of two different intermittent feeding methods on the transition to full enteral feeding in preterm infants. Study design: A prospective, randomized controlled study was conducted in a neonatology and perinatology center. Subjects: Preterm infants with a gestational age between 24 + 0/7 and 31 + 6/7 were included in this study. They were divided into two groups: the SIF (slow infusion feeding) group and the IBF (intermittent bolus feeding) group. In the SIF group, feed volumes were administered over one hour using an infusion pump through an orogastric tube, with feeding occurring every three hours. The IBF group received enteral feeding using a gravity-based technique with a syringe through an orogastric tube, completed within 10 to 30 min. Outcome measures: The primary outcome was the achievement of full enteral feeding and the occurrence of feeding intolerance. Results: A total of 103 infants were enrolled in the study (50 in SIF and 53 in IBF). The time to achieve full enteral feeding did not differ significantly between the two groups (*p* = 0.20). The SIF group had significantly fewer occurrences in which gastric residual volume exceeded 50% (*p* = 0.01). Moreover, the SIF group had a significantly shorter duration of non-per-oral (NPO) status than the IBF group (*p* = 0.03). Conclusions: It is our contention that the use of the SIF method as an alternative feeding method is appropriate for infants with feeding intolerance and those at high risk of feeding intolerance.

## 1. Introduction

As the survival rates of preterm infants increase, there is also a corresponding rise in morbidity rates. Ensuring adequate enteral nutrition and growth in surviving preterm infants is a critical milestone in neonatal intensive care units (NICUs).

According to the World Health Organization report, an estimated 13.4 million babies were born preterm in 2020, and nearly 1 million of them died due to preterm complications [1]. In low-income countries, only one out of every ten preterm babies born before 28 weeks survives, while in high-income countries, this number increases to nine out of ten [1].

Due to the lack of consensus in the literature regarding the definition of feeding intolerance and gastric residual volume, clinics generally aim to implement their own feeding protocols to achieve optimal and early enteral nutrition. Feeding intolerance is mostly a sign of immature gastrointestinal function, but it is also one of the first manifestations of some conditions such as sepsis, necrotizing enterocolitis (NEC), and spontaneous intestinal perforation, which can cause serious harm to preterm infants [2]. Based on all these, feeding intolerance leads to interruptions in the planned enteral feeding schedule and delays the time to achieve full enteral feeding [2].

Many preterm infants require enteral feeding as they cannot coordinate sucking, swallowing, and breathing. In enteral feeding, milk feeds are delivered through a small feeding tube passed via the nose or mouth into the stomach. Various enteral feeding techniques, such as intermittent or continuous feeding, push, or gravity-assisted methods, and short or long feeding intervals, are utilized to enhance feeding tolerance in preterm infants [1,2]. Tube feeding can be administered intermittently as bolus feeding through push or gravity-assisted methods, or as intermittent infusion feeding with resting intervals [3,4]. Many clinicians prefer intermittent feeding due to its physiological benefits, including the stimulation of cyclic secretion of gastrin, insulin, and gastric inhibitory peptides, thereby promoting the development of the gastrointestinal system [5,6]. Intermittent slow feeding with an infusion pump leads to the relaxation of the gastric antrum and decreased duodenal motility, particularly in infants with very low birth weight, resulting in near-complete gastric emptying. However, this method has certain drawbacks, such as difficulties in maintaining the temperature of the slowly infused feed and the requirement for an infusion pump.

However, there is no consensus on which method is most appropriate and safe for promoting enteral feeding [6].

In our clinic, the general approach to feeding is gravity-assisted intermittent bolus feeding (IBF). However, we have observed that slow infusion intermittent feeding (SIF) results in fewer interruptions in the feeding schedule, earlier attainment of full enteral feeding in infants with feeding intolerance, and less delay in the increase in feeding volumes. Based on these observations, we aimed to evaluate the impact of two different intermittent feeding methods on feeding intolerance in preterm infants.

## 2. Materials and Methods

This prospective, single-centered randomized study (NCT05265143) was conducted between February 2015 and January 2016. The neonatal intensive care unit is a 130-bed capacity, level III unit working as a perinatology and neonatology center. Preterm infants with a gestational age of ≤32 weeks were included in the study. Infants with major congenital and/or chromosomal anomalies, gastrointestinal anomalies, metabolic problems (renal, endocrine), severe cardiopulmonary compromise, or multiorgan failure, and those without parental consent, were excluded.

This study was approved by the local Ethics Committee and informed parental consent was obtained for each infant before enrollment.

### 2.1. Intervention

In this study, newborn infants within 24 h of birth were enrolled and stratified based on birth weight (BW). The babies were divided into three groups: 500–999 g, 1000–1499 g, and 1500–2000 g. Random assignment was then conducted to allocate the infants into either the slow infusion intermittent feeding (SIF) group or the intermittent bolus feeding (IBF) group. This random assignment was performed using computer-generated random allocation.

The feeding intervention was carefully managed throughout the study period, and the intervention would cease in specific circumstances. First and foremost, if parental consent was withdrawn at any point during the study, the participant’s involvement in the feeding intervention was terminated. Additionally, in cases where the occurrence of necrotizing enterocolitis (NEC) reached stage II or higher, or if mortality was observed, the feeding intervention was promptly discontinued for the affected patients.

Throughout the study, the enteral feeding groups remained consistent, and the feeding intervention persisted until each patient successfully achieved oral feeding. This approach aimed to ensure that the enteral feeding methods were diligently followed, and the outcomes were appropriately assessed based on each patient’s progress.

Moreover, the follow-up of all patients was conducted diligently until the time of discharge. This meticulous follow-up allowed for a comprehensive evaluation of the impact of the feeding intervention on patients’ health and overall well-being during their stay in the neonatal care unit.

### 2.2. Feeding Protocol

In both groups, feeding was initiated within the first 24 h, and the feed volume was increased by 10–20 mL/kg/day during the first 72 h, followed by an increase of 20–25 mL/kg/day thereafter according to our unit protocol. The achievement of a daily enteral feeding volume of 150 mL/kg sustained for at least 72 h was defined as full enteral feed.

The first group, referred to as the SIF group, received feed volumes administered within one hour using an infusion pump through an orogastric tube. Feeding occurred every three hours, with one hour dedicated to feeding and two hours for resting cycles. We extended the time interval in the SIF group compared with the IBF group to allow for gastric emptying. This information is based on a study, which indicated that with a 2 h SIF interval, approximately one-fourth of gastric emptying occurs [3]. We selected a one-hour duration for this purpose to ensure the maintenance of milk temperature.

In the second group, referred to as the IBF group, enteral feeding procedures were carried out using a gravity-based technique with a syringe through an orogastric tube over a period of 10 to 30 min. The syringe, containing the prescribed feed volume, was positioned 10 cm above the infant, allowing the milk to flow passively into the gastrointestinal tract via the force of gravity.

Feeding intolerance was defined as the occurrence of bilious emesis or gastric residuals, visibly bloody stools, abdominal distention or tenderness, abdominal discoloration, gastric residuals exceeding 50% of the previous feed volume, emesis occurring three or more times within a 24 h period, or the presence of clinical or radiological evidence of necrotizing enterocolitis (NEC). Gastric residuals were assessed every 8 h prior to subsequent feeding procedures. The number of daily occurrences of gastric residuals was recorded based on nurse observation charts.

When there were signs of feeding intolerance, the amount of feeding was not increased the next day. When there was only ≥50% gastric residual volume (GRV) and no other signs of feeding intolerance, the next feeding amount was reduced by 50% for a single time. When gastric residuals developed ≥3 times during the day, all remaining feeding amounts were reduced by 50% on that day according to our unit protocol.

Abdominal X-ray imaging was performed in the presence of abdominal distension/discoloration, bilious emesis, or bloody stool. In the presence or suspicion of NEC, feeding was interrupted in both groups. The feeding of the infants with suspicious findings in the abdominal radiographs was interrupted for 24 h, and these infants were reevaluated after 24 h. If the feeding was interrupted due to feeding intolerance, and if the abdominal physical and radiographic examination was normal and the infant was clinically stable, feeding was resumed with half of the feeding volume. These interventions for feeding intolerance are already part of our clinic’s routine protocol. The primary physician of the infant decided the amount of feeding volume, the cessation of feeding, and the administration of medical treatment when feeding intolerance developed based on NICU protocols. A pediatric surgeon blinded to feeding groups also evaluated the infants for signs of necrotizing enterocolitis. In cases of bradycardia (<80/min) and critical desaturation (<80%) during feeding, feeding was discontinued, and episodes were recorded.

Our first choice for enteral nutrition was to provide breast milk. In cases where breast milk was unavailable or insufficient, preterm formula was administered. It was not a routine practice in our unit to use donor milk. As per our unit’s protocol, parenteral nutrition was initiated within the first hours of hospitalization. The discontinuation of lipid solutions occurred when the amount of enteral nutrition reached 80 mL/kg/day, and the complete cessation of parenteral nutrition was achieved when the amount reached 100 mL/kg/day. Human milk fortifiers (Eoprotin, Nutriceal Foods, SA, Santarem, Portugal) were added to breast milk once the enteral feeding volume reached 100 mL/kg/day.

### 2.3. Outcome Parameters

The primary outcome of our study was the achievement of full enteral feeding and the occurrence of feeding intolerance, including the number of non-per-oral (NPO) days. Data pertaining to feeding tolerance, including gastric residuals, the number of apnea episodes, and the number of feeding interruptions were collected on a daily basis until the achievement of full oral feeding. Secondary outcomes included the number of days required to regain birth weight, the incidence of late-onset sepsis, and the incidence of bradycardia events during feeding.

### 2.4. Sample Size

On the basis of our previous data from our NICU, we hypothesized that intermittent feeding would reduce the rate of feeding intolerance to 35% (absolute reduction of 30%) with a two-sided alpha error of 0.05 and beta error of 0.2, with 80 % power; the estimated sample size was 98 (49 infants in each group).

### 2.5. Statistical Analysis

The frequency and percentage distributions related to the data are presented. The relationship between variables measured at a categorical level was evaluated using the Chi-square test. Whether the data showed normal distribution was examined with the Kolmogorov–Smirnov test, and an independent *t*-test was used for the data showing a normal distribution. The significance level was accepted as *p* < 0.05. All statistical analyses were performed using the SPSS for Windows software version 17.0 (SPSS Inc., Chicago, IL, USA).

## 3. Results

During the study period, a total of 110 infants were enrolled in the study. Four patients declined to give consent for participation, two patients were transferred to another hospital, and one patient developed stage III necrotizing enterocolitis, as shown in the flowchart. (Figure 1). The demographic and clinical characteristics of the two groups are listed in Table 1.

The mean gestational age and birth weight were 29.5 ± 1.9 weeks and 1278 ± 255 g in the SIF group and 29.1 ± 1.8 weeks and 1203 ± 212 g in the IBF group (*p* > 0.05 for both variables).

The results of the primary and secondary outcomes are presented in Table 2. The time to achieve full enteral feeding was similar among the two groups (13.6 ± 8.05 vs. 16.2 ± 12.3 *p* = 0.20). The number of occurrences in which gastric residual volume exceeded 50% was significantly lower in the SIF group (1.12 ± 1.56 vs. 2.15 ± 1.4, *p* = 0.01). Additionally, the SIF group had significantly fewer days on the NPO status (1.9 ± 2.7 vs. 4.2 ± 6.6, *p* = 0.03). The time to gain birth weight was significantly shorter in the SIF group (10.8 ± 3.2 versus 12.7 ± 4.17; *p* = 0.01). The incidence of apnea and bradycardia episodes was similar in both groups (4% vs. 5.7% and 6% vs. 3.8%, respectively). The rates of late-onset sepsis were also found to be similar in both groups (28% vs. 28.3%).

## 4. Discussion

In this study, we compared two different intermittent feeding methods in preterm infants that, to the best of our knowledge, have not been compared so far. We observed that the incidence of feeding intolerance was significantly lower with a SIF strategy. The infants fed with SIF reached their birth weight earlier; however, there was no difference in days to achieve full enteral feeding among the two feeding strategies.

In the literature, the most similar study to our own was conducted by Kumar et al., comparing continuous feeding, intermittent bolus infusion, and intermittent bolus gravity feeding [7]. In their study, Kumar et al. did not find any significant differences in feeding intolerance and achievement of full enteral feeding among the three feeding models. However, in our study, we attributed the lower incidence of feeding intolerance in the SIF group to the longer infusion duration. In SIF, the duodenal motor response is similar to that in adults, and gastric emptying is faster [3]. The slow infusion method also has the advantage of a longer exposure time for absorption with smaller volumes [3]. This may explain the lower incidence of feeding intolerance in the SIF group in this study. We also observed that feeding with SIF was well tolerated in preterm infants without causing any side effects such as apnea or bradycardia.

Previous research demonstrated that continuous feeding promotes faster growth and full enteral feeding time, especially in infants with extremely low birth weight [8,9]. However, a meta-analysis of randomized trials revealed no difference in the incidence of feeding intolerance, days to reach full enteral feeding, or days to regain birth weight between intermittent and continuous feeding [5]. In fact, intermittent feeding was found to be more advantageous in terms of time to reach full enteral feeds. But the number of infants included was very small; birth weights were higher; and feeding protocols, including the timing of first enteral feeding, minimal enteral feeding, milk composition, and volume of daily increments, were diverse. In a study conducted by RövekampAbels et al., a comparison was performed between semi-continuous feeding and IBF in preterm infants, revealing that IBF was associated with reduced feeding intolerance [10].

IBF is preferred as a more physiological feeding method in preterm infants since it has been shown to stimulate cyclic hormone release and protein synthesis in previous animal and human studies [11,12,13]. Continuous feeding is a form of feeding with drip infusion over 24 h. It has the advantage of better nutrient absorption and lower energy expenditure [14,15]. In the present study, both feeding methods were intermittent, but one of the groups was fed similar to continuous feeding, with drip infusion for a longer time compared with bolus feeding. In this way, the negative effects of lower and non-pulsatile insulin and gastric hormone secretion, low amino acid accumulation, and protein synthesis that may occur in continuous nutrition, as well as glycemic fluctuations and metabolic and respiratory instability that may occur in intermittent bolus nutrition, can be avoided.

GRV is commonly used as an indicator of feeding intolerance and a warning sign for necrotizing enterocolitis in premature infants [16,17]. It is believed that gastric residuals are likely innocent consequences resulting from delayed gut maturation and motility in low-birth-weight infants, and there are no established normal standards for GRV levels. However, the literature lacks a consensus regarding the specific threshold of GRV that definitively indicates feeding intolerance [18,19].

In our study, we carefully assessed both GRV measurements and the clinician’s decision to withhold feeding (NPO) as indicators of feeding intolerance. Interestingly, both parameters were found to be significantly lower in the SIF group than in the IBF group. These findings suggest that a selective approach in evaluating and managing GRV, rather than adhering to strict and universally defined thresholds, might be beneficial in optimizing the feeding process and reducing feeding intolerance in premature infants. Further research is warranted to establish evidence-based guidelines for managing GRV in this vulnerable population, taking into account the potential implications for NEC risk and overall feeding success.

In addition to the fact that IBF is more physiological, another reason for its preference is that it is easy to apply. It has the advantages of easy preparation and a shorter nursing time. Since it is given in a short time, feeding is finished without causing a change in the temperature of the breast milk or formula. No additional material is needed. In SIF, the temperature of formula or breast milk may change as the duration of delivery is prolonged, and the fat and other nutritive particles may precipitate into the infusion set. An infusion pump and infusion set are required as additional devices for nutrition. This additional economic cost is a disadvantage for SIF.

Signs of feeding intolerance such as abdominal distension, slow motility, and increased gastric residuals are similar to earlier signs of NEC [18,19]. Although NEC is a devastating disease, thankfully, it is rather rarely observed, but feeding intolerance due to gastrointestinal immaturity is a common condition that affects one out of four preterm infants [20]. Distinguishing between feeding intolerance due to gastrointestinal immaturity and the clinical relevance of other conditions like sepsis and NEC, which present themselves as feeding intolerance, leads to unnecessary blood sampling, X-ray examinations, and feeding interruptions. Moreover, recent research demonstrated that feeding intolerance alters the gut microbiota of preterm infants, increasing the growth of Klebsiella [21].

Intestinal microbiota diversity is associated with improved feeding tolerance and growth [22]. We know the importance of the continuity of enteral feeding for intestinal adaptation and immune function in preterm infants [23,24].

Early, adequate, and safe full enteral nutrition and optimal growth of the preterm are the major goals of neonatal care. However, the current level of evidence does not support the superiority of one feeding method over another [6].

Our study has several strengths that contribute to its significance and novelty in the literature. Firstly, to the best of our knowledge, there has been a lack of previous research comparing these two specific feeding methods in the context of our target population, making our investigation one of the first of its kind. This aspect of originality enhances the relevance of our findings and potential contributions to the field of neonatology.

Moreover, an important aspect of our study’s strength lies in the successful attainment of our desired sample size. Adequate sample size is crucial for ensuring the statistical power and generalizability of the study results. By reaching our target sample size, we have increased the reliability and robustness of our conclusions, enhancing the credibility of our findings.

However, despite these strengths, there are also certain limitations to be acknowledged. One primary limitation is the inability to implement the blinding of the feeding method due to the nature of the study design. A single-center design, lack of consensus in the literature regarding the definition of feeding intolerance, and variations in opinions regarding the measurement of GRV can be considered the other limitations of the study.

Despite these limitations, our study provides valuable insights into the field of neonatal nutrition, and future research may build upon these findings to further elucidate the optimal feeding approach for our target population.

Considering these factors, achieving feeding tolerance and preventing feeding interruptions are crucial to promoting optimal growth and development in preterm infants. Despite the identified disadvantages, we believe that SIF is a better-tolerated feeding method, especially for preterm infants with feeding intolerance or at risk for feeding intolerance. In these cases, SIF may be preferred over another feeding method, IBF, until the maturation of the gastric system is achieved.

## 5. Conclusions

Our study contributes valuable insights to the field of neonatal nutrition, despite certain limitations such as the inability to implement blinding and the exclusion of additional measurements. We are confident that future research can build upon these findings to further elucidate the optimal feeding approach for our target population. The pursuit of early, adequate, and safe full enteral nutrition remains a primary goal in neonatal care, and evidence-based practices will continue to shape the best approach for achieving this goal. We concluded that it would be more beneficial for each infant to be evaluated individually and if the enteral feeding method is specific to the infant rather than conducted as a clinical practice routine.

## Figures and Tables

**Figure 1 children-10-01389-f001:**
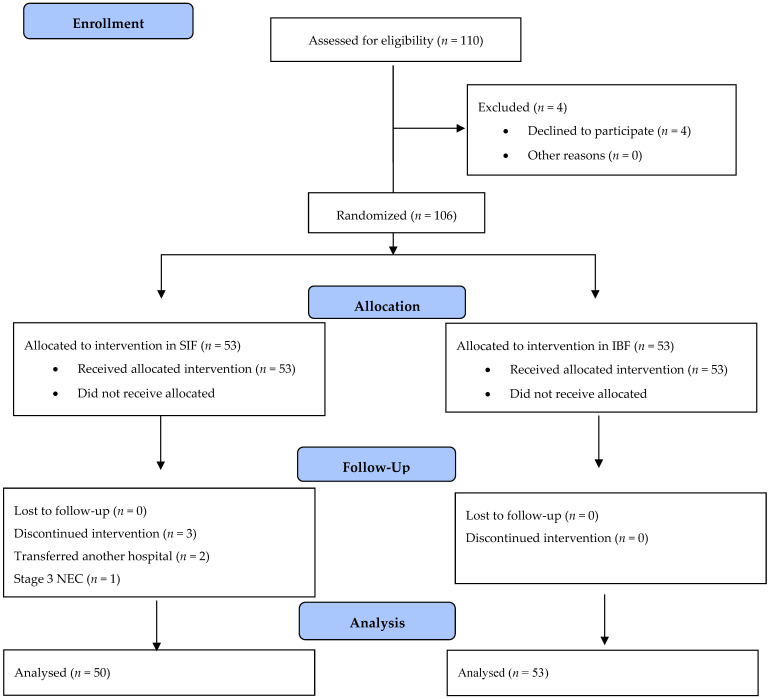
Study flowchart.

**Table 1 children-10-01389-t001:** Demographic and clinical characteristics.

	SIF Group (*n*: 50)	IBF Group (*n*: 53)	*p*
Birth weight, g	1278 ± 255	1203 ± 212	0.11
Gestation, wk	29.5 ± 1.9	29.1 ± 1.8	0.25
Males	24 (48)	25 (47)	1.00
SGA	9 (18)	10 (18.9)	1.00
PE/HELLP	10 (20)	14 (26)	0.50
Antenatal steroids	34 (68)	31 (58.5)	0.41
Cesarean section	45 (90)	45 (84.9)	0.55
Apgar (5 min)	8 (3–9)	8 (3–9)	0.29
Multiple births	11 (22)	12 (22.6)	1.0
PDA	11 (22)	19 (35.8)	0.13
Intraventricular hemorrhage, grade > 2	3 (6)	1 (1.9)	0.35
Feeding with OMM	39 (78)	38 (71.7)	0.47
CRIB score	2 (1–8)	4 (1–14)	0.15

SIF: slow infusion intermittent feeding, IBF: intermittent bolus feeding, BW: birth weight, CRIB: Clinical Risk Index for Babies; HELLP: hemolysis elevated liver enzymes and low platelets; IQR: interquartile range; PDA: patent ductus arteriosus; PE: preeclampsia; OMM: own mother milk; SDS: standard deviation score; SGA: small-for-gestational age. Data are represented as mean ± SD, median IQR, or count %.

**Table 2 children-10-01389-t002:** Primary and secondary outcomes of infants.

Primary Outcome	SIF Group (*n*: 50)	IBF Group (*n*: 53)	*p*
Days to achieve full enteral feeding/day	13.6 ± 8.05	16.2 ± 12.3	0.20
GRV > 50%, *n*	1.12 ± 1.56	2.15 ± 1.4	0.01
Days of NPO, d	1.9 ± 2.7	4.2 ± 6.6	0.03
**Secondary Outcomes**			
Time to regain BW/day	10.8 ± 3.2	12.7 ± 4.17	0.01
Apnea episodes during feeding	2 (4)	3 (5.7)	1.0
Bradycardia episodes during feeding	3 (6)	2 (3.8)	0.67
LOS	14 (28)	15 (28.3)	1.0

LOS: late-onset sepsis: positive blood culture results and/or met specific criteria such as elevated C-reactive protein levels, a total leukocyte count > 25,000/mm^3^ or <5000/mm^3^, an immature/total neutrophil ratio > 0.2, or a band count > 10%. GRV: gastric residual volume, which refers to the volume of gastric contents remaining in the stomach, was measured every 3 h throughout the hospitalization period. The GRV values are expressed as a percentage of the total number of measurements performed by the nursing staff. The frequency of GRV measurements was consistent across all groups. NPO: non-per oral. Data are represented as mean ± SD, median IQR. or count %.

## Data Availability

Data is contained within the article.

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
