# Peer review of "A Comparison of Slow Infusion Intermittent Feeding versus Gravity Feeding in Preterm Infants: A Randomized Controlled Trial"

_children, 2023, doi:10.3390/children10081389_

Round 1

Reviewer 1 Report

Thank you for submitting the manuscript "Comparison of slow infusion intermittent feeding versus gravity feeding in preterm infants: A randomized controlled trial" to Children. The subject of the manuscript is interesting and has a gap in the literature with sensitive information.

Line#19: instead of saying that the gestational age was less than 32 weeks, I think it would be more interesting to present the range or average weeks of preterm infants.

Line#29: NPO? first time this abbreviation appears in the abstract.

Line#46: also describe the advantages/disadvantages of using slow infusion.

Overall, the introduction needs to bring more arguments to justify the study (although I understand the justification). A reader would like more data related to the central problem of the study, for example, how many preterm infants are born in a year? How many of these end up being affected by gastrointestinal problems in the short and long term, etc.

Line#66: stratified how? include information about it.

Line#70: Will you talk about the percentage of the sample that resulted in NEC later in the results item? If not, please consider including it here.

Line#81: I thought this feeding interval was long because newborns have a very fast metabolism and during pregnancy they don't have "time to eat" with intervals. Please justify what this time was based on.

Line#98: But was this feed intended to reach how much of the recommended percentage? I ask this because there are several studies, including with other patient age groups, which have shown that there is a big difference between the recommended diet and the administered diet when talking about enteral nutrition.

Line#112: if it is not a value that will be presented later, it is interesting to include here the percentage of patients who received breast milk or formula, etc. Even if this seems indifferent to the type of feeding mechanism, it is decisive in the presence of gastrointestinal symptoms.

The English language is fine.

Author Response

Point 1: Line#19: instead of saying that the gestational age was less than 32 weeks, I think it would be more interesting to present the range or average weeks of preterm infants.

Respond: I made the necessary revisions following your suggestions

 Point 2: Line#29: NPO? first time this abbreviation appears in the abstract.

Respond: I made the necessary revisions following your suggestions

 Point 3: Line#46: also describe the advantages/disadvantages of using slow infusion.

Overall, the introduction needs to bring more arguments to justify the study (although I understand the justification). A reader would like more data related to the central problem of the study, for example, how many preterm infants are born in a year? How many of these end up being affected by gastrointestinal problems in the short and long term, etc.

Respond: I have taken your suggestions into account and expanded the introduction section accordingly.

 Point 4: Line#66: stratified how? include information about it.

Respond: I have made the necessary additions to the methods section.

Point 5: Line#70: Will you talk about the percentage of the sample that resulted in NEC later in the results item? If not, please consider including it here.

Respond: In the study flow chart, there is data related to this issue. However, we decided not to include it in the manuscript because one patient from the SIF group was excluded from the study due to stage 3 NEC (Necrotizing Enterocolitis). On the other hand, there were no patients with advanced stage NEC in the IBF group.

Point 6: Line#81: I thought this feeding interval was long because newborns have a very fast metabolism and during pregnancy they don't have "time to eat" with intervals. Please justify what this time was based on.

Respond: We have now added the supporting data for the feeding duration in the SIF group to the methods section. As for the IBF group, the feeding duration aligns with the standard feeding time recommended by the World Health Organization.

Point 7: Line#98: But was this feed intended to reach how much of the recommended percentage? I ask this because there are several studies, including with other patient age groups, which have shown that there is a big difference between the recommended diet and the administered diet when talking about enteral nutrition.

Respond: Our aim was to provide at least 50% of the feeding volume at each feeding session

 Point 8: Line#112: if it is not a value that will be presented later, it is interesting to include here the percentage of patients who received breast milk or formula, etc. Even if this seems indifferent to the type of feeding mechanism, it is decisive in the presence of gastrointestinal symptoms.

Respond: We would like to clarify that we have now included information regarding the lack of difference in feeding contents between the two groups in the demographic data table. We sincerely appreciate your attention and guidance throughout the review process.

Reviewer 2 Report

This paper deals with an important clinical problem in neonatology. Two different oral feeding regimes in preterm infants were compared in a randomized controlled trial.

The hypothesis is well stated and the number of patients needed calculated. The two feeding regimes are adequately described as well as the criteria for patient selection, exclusion and primary and secondary outcome parameters.

The results are clearly presented and the discussion sound.

My only critical comment concerns the conclusions, both in the abstract and the main paper: Firstly, they do not match and secondly, they are really vague and hardy reflect the results. I would prefer a more specific conclusion, similar to the statement given in the last paragraph of the discussion.   

Author Response

Point 1: My only critical comment concerns the conclusions, both in the abstract and the main paper: Firstly, they do not match and secondly, they are really vague and hardy reflect the results. I would prefer a more specific conclusion, similar to the statement given in the last paragraph of the discussion.   

Respond: We made the necessary revisions following your suggestions

Reviewer 3 Report

This is a well-designed and well-written randomized controlled trial study. The authors compared the time to achieve full enteral feeding in preterm infants between the slow infusion intermittent feeding group and the intermittent bolus feeding group, providing evidence for the development of dietary plans for preterm infants. Although the results showed no statistically significant difference in the time to achieve full enteral feeding between the two groups, the slow infusion intermittent feeding group had a lower number of gastric residual volume (GRV) > 50% and fewer days of non-per-oral feeding (NPO).

Despite the authors' thorough discussion of the results, it is recommended that they further elaborate on the discussion of GRV and NPO, considering that the difference in the primary outcome was not statistically significant, and also highlight the innovation of this study. This may be helpful in indicating the quality of the article.

Author Response

Point1: Despite the authors' thorough discussion of the results, it is recommended that they further elaborate on the discussion of GRV and NPO, considering that the difference in the primary outcome was not statistically significant, and also highlight the innovation of this study. This may be helpful in indicating the quality of the article.

Respond: We made the necessary revisions following your suggestions